# Mechanisms and Therapeutic Effects of Benzoquinone Ring Analogs in Primary CoQ Deficiencies

**DOI:** 10.3390/antiox11040665

**Published:** 2022-03-30

**Authors:** Alba Pesini, Agustin Hidalgo-Gutierrez, Catarina M. Quinzii

**Affiliations:** Department of Neurology, Columbia University Medical Center, New York, NY 10032, USA; ap4072@cumc.columbia.edu (A.P.); ah3591@cumc.columbia.edu (A.H.-G.)

**Keywords:** coenzyme Q_10_, analogs, 4-hydroxybenzoic acid, mitochondria, preclinical models

## Abstract

Coenzyme Q (CoQ) is a conserved polyprenylated lipid composed of a redox-active benzoquinone ring and a long polyisoprenyl tail that serves as a membrane anchor. CoQ biosynthesis involves multiple steps, including multiple modifications of the precursor ring 4-hydroxybenzoic acid. Mutations in the enzymes involved in CoQ biosynthesis pathway result in primary coenzyme Q deficiencies, mitochondrial disorders whose clinical heterogenicity reflects the multiple biological function of CoQ. Patients with these disorders do not always respond to CoQ supplementation, and CoQ analogs have not been successful as alternative approaches. Progress made in understanding the CoQ biosynthesis pathway and studies of supplementation with 4-hydroxybenzoic acid ring analogs have opened a new area in the field of primary CoQ deficiencies treatment. Here, we will review these studies, focusing on efficacy of the different 4-hydroxybenzoic acid ring analogs, models in which they have been tested, and their mechanisms of action. Understanding how these compounds ameliorate biochemical, molecular, and/or clinical phenotypes of CoQ deficiencies is important to develop the most rational treatment for CoQ deficient patients, depending on their molecular defects.

## 1. Introduction

Coenzyme Q (CoQ) is a lipid with antioxidant and electron carrier capacities [1]. It was isolated and characterized by Festenstein et al. in 1955 [2] and soon after established as a compound involved in the mitochondrial respiratory chain by Crane et al. [1,3]. Since then, CoQ has been a widely studied molecule, and numerous studies have shown its roles in a variety of cellular processes [1]. For example, it functions as a cofactor for dihydroorotate dehydrogenase (DHODH) in pyrimidine biosynthesis; sulfide-quinone oxidoreductase (SQOR) in sulfide oxidation; mitochondrial glycerol-3-phosphate dehydrogenase (G3PDH), that links glycolysis, oxidative phosphorylation, and fatty acid metabolism; electron transport flavoprotein dehydrogenase (ETFDH), involved in fatty acid β-oxidation, catabolism of several amino acids, and sarcosine metabolism; proline dehydrogenase (PRODH) and proline dehydrogenase 2 (PRODH2) required for proline, glyoxylate, and arginine metabolism; and choline dehydrogenase (CHDH) that is related to glycine metabolism [4,5]. CoQ is also required for the maintenance of the lysosomal lumen acidity, necessary for the correct functioning of this organelle [6]. Thus, not surprisingly, defects in CoQ biosynthesis cause CoQ deficiency associated with a variety of human diseases [7].

Although CoQ deficiencies are potentially treatable, not all patients respond to CoQ supplementation [8]. The low bioavailability of CoQ and its inability to cross the blood–brain barrier (BBB) indicates the need for alternative therapeutic approaches [9], especially in patients with neurological involvement [10]. Some of these alternatives include supplementation with (1) CoQ reduced form (ubiquinol), instead of the oxidized one (ubiquinone) [11]; (2) water-soluble CoQ formulation, which has already shown an increase in the bioavailability [12,13]; and (3) CoQ analogs as Idebenone, mitoquinone, and vatiquinone, which have been only partially successful [10].

More recently, 4-HB analogs have emerged as potential therapeutic agents. These compounds are hypothesized to rescue CoQ production, bypassing molecular defects in different steps of the CoQ synthesis pathway. However, in vitro and in vivo studies in a variety of models have revealed that their mechanisms are more complex and require further elucidation before they can be implemented in the clinical practice.

## 2. CoQ Biosynthesis

CoQ biosynthesis is a complex mechanism that has been investigated in different species, mostly in bacteria and yeast [14,15]. To date, up to 14 genes that encode proteins involved in the CoQ synthesis have been identified in yeast [14]. CoQ synthesis mainly takes place at the mitochondrial inner membrane. However, some studies have shown CoQ synthesis outside the mitochondria, specifically in the Golgi apparatus and the endoplasmatic reticulum (ER) [16,17]. In addition, recent studies in yeast have pointed out the role of the ER-mitochondria structure (ERMES) in CoQ biosynthesis, underlying the relevance of the communication between these two organelles [18,19]. The synthesis of CoQ is similar in prokaryotes and eukaryotes: a long polyisoprenoid lipid tail is initially coupled to a benzenoid precursor, and the benzenoid ring is further modified through successive steps to yield the final product [20,21,22]. The universal benzenoid precursor ring is 4-hydroxybenzoic acid (4-HB), although other alternative ring precursors have been proposed [20].

The isoprene carbon units for the synthesis of CoQ side chain are derived from the mevalonate pathway in eukaryotes and some prokaryotes [20,23]. The number of isoprene units of the polyisoprenoid chain is determined by a species-specific polyprenyl diphosphate synthase (IspB in *Escherichia coli* (*E.coli*) Coq1 in *Saccharomyces cerevisiae* (*S.cerevisiae*), and PDSS1-PDSS2 in mammals). Each species has a major CoQ form: 6 isoprenyl units in *S. cerevisiae*, 8 units in *E. coli*, 9 units in mice, and 10 units in humans. This polyisoprenoid tail is attached to the position C-3 of the ring by UbiA/Coq2/COQ2 (*E. coli*, *S. cerevisiae*, and mammals, respectively) (Figure 1) [14,20,24,25,26]. Then, the ring (3-hexaprenyl-4HB; HBB) is modified to produce the final benzoquinone ring of CoQ; specifically, Coq6/COQ6 catalyzes the C-5 hydroxylation; UbiG/Coq3/COQ3 catalyzes the C-5 O-methylation; C1-decarboxylation and C1-hydroxylation enzyme(s) have not been identified; UbiE/Coq5/COQ5 catalyzes the subsequent C-2 methylation; UbiF/Coq7/COQ7 the C-6 hydroxylation; and in the final step, UbiG/Coq3/COQ3 catalyzes the C-6 *O*-methylation to produce CoQ [20].

Other regulatory proteins are involved in CoQ biosynthesis [20,23]. In eukaryotes, they include Coq4/COQ4, which has been hypothesized to act as a scaffolding protein that binds proteins and lipids [27]; Coq8/COQ8A-B, which are kinases that phosphorylate other COQ proteins to stabilize the Q protein complex [28,29]; Coq9/COQ9, which interacts with COQ7 and is essential for its stability and activity [30,31]; and Coq10/COQ10 and Coq11/COQ11 that have been shown to be involved in CoQ biosynthesis and transport to the mitochondrial respiratory chain, but whose exact functions remain unknown [32,33]. Some of these proteins are organized to form a multiprotein complex called the Q complex that appears to be essential for CoQ synthesis [20,23]. To date, defects in PDSS1, PDSS2, COQ2, COQ4, COQ5, COQ6, COQ7, COQ8A, COQ8B, and COQ9 have been found to cause human CoQ deficiency and disease [34].

## 3. 4-Hydroxybenzoic Acid Analogs

4-HB is the universal benzoquinone ring precursor for CoQ. Its synthesis varies depending on the specie. In *E. coli*, 4-HB is synthesized by UbiC that catalyzes a 3horismite pyruvate-lyase reaction [35,36]. It uses chorismic acid as a substrate, an intermediate of the shikimate pathway involved in the biosynthesis of aromatic amino acids [37]. In *S. cerevisiae*, more than one step is needed. 4-hydroxyphenyl pyruvate (4-HPP) is obtained from the shikimate pathway or from exogenous tyrosine. 4-HPP is further converted to 4-hydroxybenzaldehyde (4-Hbz) via uncharacterized steps [15], and as a final reaction, 4-Hbz is oxidized to 4-HB by the aldehyde dehydrogenase Hfd1 [15,23]. Mammals do not possess the shikimate pathway. 4-HB derives from tyrosine and phenylalanine [38,39], via a pathway that is still not well characterized [40]. The last reaction, the oxidation of 4-Hbz to 4-HB, was recently discovered in *S. cerevisiae* [15,23,40].

Supplementation with 4-HB has been shown to rescue CoQ levels in CoQ-deficient bacteria (*Xanthomona Oryzae*), plants (*Arabidopsis*), and yeast (*S. cerevisae*) with defects in XanB2, C4H, and Hfd1, respectively, that are involved in the 4-HB ring synthesis [15,23,40,41,42]. Furthermore, 4-HB supplementation in CoQ-deficient fibroblasts from three different patients carrying *COQ2* mutations was able to normalize CoQ levels [43]. The normalization of CoQ levels due to the administration of COQ2 substrate suggested that bypassing a step of the CoQ biosynthesis pathway could be an effective therapy in CoQ deficiency.

### 3.1. 3,4-Dihydroxybenzoic Acid and Vanillic Acid

In 1977, Nambudiri et al. studied the hydroxylation of 4-HB in mammals [44]. They showed that 4-HB was hydroxylated to polyprenyl-hydroxybenzoate (PPHB) but also to a PPHB methylated derivate, vanillic acid (VA). The study showed also that rat mitochondria were able to prenylate VA and 3,4-dihydroxybenzoic acid (3,4-diHB) and introduced them in the CoQ synthesis pathway. A year later, in 1978, Goewert et al. showed that prenylated VA accumulated in a *S. cerevisiae* model unable to synthesize CoQ_6_ [45]. 3,4-diHB is a benzoquinone ring already hydroxylated in the position C3, a reaction that in CoQ biosynthesis is catalyzed by COQ6, while VA possess the *O*-methylation in the position C3, in CoQ biosynthesis catalyzed by COQ3. Thus, these molecules not only already carry the chemical modification necessary for the benzoquinone ring to produce the final product of CoQ biosynthesis, but they can also enter in the CoQ biosynthetic pathway through COQ2 and bypass defective modification steps.

Since the publication of the Namburidi and Goewert work, VA and 3,4-diHB, have been used in different CoQ deficiency models. Both analogs were able to rescue CoQ_6_ biosynthesis and respiration in a *S. cerevisiae Coq6* mutant; however, the effects of the 4-HB analogs were dependent upon overexpression of Coq8, which was previously shown to stabilize the Q complex [29,46,47]. Both compounds, especially VA, were also able to restore growth in another *S. cerevisiae* model carrying a *Coq6* human mutation [48]. In addition, surprisingly, VA supplementation partly restored CoQ biosynthesis in *Drosophila* garland cell nephrocytes (GCN) a model of human COQ2 nephropathy [49], indicating that VA efficacy in CoQ deficiency was not due entirely to its ability to bypass a dysfunctional step of CoQ biosynthesis, as further discussed in the next section.

VA stimulated CoQ_10_ biosynthesis and improved cell viability in patient-derived fibroblasts with a genetic defect in *COQ9* [43]. Furthermore, VA treatment restored CoQ_10_ biosynthesis and, consequently, ROS levels, cellular respiration, and ATP production in a *COQ6*-depleted HeLa cells, where the supplementation with CoQ_10_ was only partially effective [50]. Additionally, VA but not 3,4-diHB, improved the pathological podocytes migration pattern in COQ6-depleted human podocytes [51].

Overall, VA seems to be more effective than 3,4-diHB in CoQ deficiency. Moreover, VA has several advantages as a potential therapeutic approach in patients: (1) In vivo, it can be produced in the liver through oxidation of vanillin [52,53]; (2) it is considered non-toxic and safe for human use by the FDA, as it is commonly used as a flavoring agent [54]; and (3) it has good bioavailability and can cross the blood–brain barrier efficiently [54], overcoming one of the major limitations of oral CoQ supplementation [50].

### 3.2. β-Resorcylic Acid

The β-resorcylic acid (β-RA), also named 2,4-dihydroxybenzoic acid, is a benzoquinone ring that was already hydroxylated in C2 position. This hydroxylation is naturally catalyzed by COQ7 protein in one of the last steps of CoQ biosynthesis [55].

Like VA, β-RA has been widely used in the food industry; because of its ability to enhance the sweetness of other artificial sweeteners, it is added as a food flavor modifier in beverages, fish products, snack foods, and chewing gums [56]. Along with VA and 3,4-diHB, β-RA was tested in several models of CoQ deficiency (Figure 2).

In a *Coq7* knockout *S. cerevisiae* model, β-RA supplementation has been shown to restore CoQ_6_ biosynthesis, after Coq8 overexpression [47,57]. By comparing β-RA and 4-HB supplementation, Xie et al. showed that 4-HB was unable to rescue CoQ_6_ biosynthesis and to decrease the accumulation of the toxic metabolite DMQ_6_ [47]. DMQ is the intermediate used by COQ7 throughout the biosynthesis of CoQ and has been shown to inhibit complexes I+III activities of the mitochondrial respiratory chain in *Caenorhabditis elegans* (*C. elegans*) [58].

β-RA supplementation was shown to rescue CoQ_10_ biosynthesis and mitochondrial respiration in CoQ_10_-deficient fibroblasts from patients with *COQ7* mutations [59]. Interestingly, β-RA supplementation produced different effects in human fibroblast cell lines with two different COQ7 mutations, V141E and L111P. The V141E mutation causes severe CoQ deficiency associated with severe multisystemic disease [59], while the L111P mutation causes a milder decrease in CoQ and increase in DMQ levels, and less severe phenotype than the V141E mutation [21]. β-RA administration decreased DMQ accumulation in both cell lines, but significantly increased CoQ_10_ levels only in the line carrying the V141E mutation, indicating that β-RA affects CoQ_10_ synthesis but has different effects, depending on the extent of the loss in enzyme activity [21]. These beneficial effects of β-RA were reproduced in other COQ7 and COQ9 mutant fibroblasts from different patients [43,60].

β-RA supplementation increased CoQ_9_ levels also in mouse Coq7-depleted fibroblasts [56] and, when added to drinking water, was able to partially rescue the phenotype in *Coq7* KO mice, increasing their weight and life span [21,56]. Amelioration of the clinical phenotype was associated with partial rescue of the levels of mitochondrial respiration defect in the kidney, lactate, and triglycerides levels in blood [56]. CoQ_9_ levels were increased, while accumulation of DMQ_9_ was decreased in heart, kidneys, and skeletal muscle [56]. This decrease in DMQ_9_ indicates that β-RA supplementation causes impairment of the endogenous CoQ biosynthetic pathway by competing with 4-HB, which normally acts as precursor of the CoQ benzoquinone ring [40,56,61].

Abnormalities of CoQ and DMQ levels similar to what was observed in *Coq7* mutant mice were reported also in *C. elegans* with mutations in clk-1 (COQ7 in humans) [62], human *COQ9* mutant fibroblast [60], and *Coq9^R239X^* mice [60,61].

β-RA supplementation restored survival and phenotype of *Coq9^R239X^* mutant mice, improved mitochondrial bioenergetics, and increased CoQ_9_ levels in kidneys, and decreased DMQ_9_ levels in kidney, liver, skeletal muscle, and heart [5,60,61]. Interestingly, β-RA supplementation decreased molecular, histopathological, and clinical signs of mitochondrial encephalopathy associated with CoQ_9_ deficiency in these *Coq9^R239X^* mice, without improving CoQ_9_ and DMQ_9_ levels, and mitochondrial bioenergetics in their brain [61]. Although the mechanisms by which β-RA supplementation rescues the phenotype of *Coq9* mutant mice remain to be elucidated, all these data together support the critical role of DMQ/CoQ ratio in CoQ biosynthesis [5,61]. Furthermore, studies of β-RA supplementation in *Coq9* mutant mice confirm the different effects of this compound on CoQ_9_ biosynthesis observed in the experiments in *COQ7* mutant cells with different molecular defects. In fact, contrary to what was shown in *Coq9^R239X^*, β-RA supplementation decreased CoQ_9_ levels in kidney of *Coq9^Q95X^* mice, another COQ9 mutant mouse, which manifested with mild late-onset mitochondrial myopathy, associated with moderate CoQ deficiency [60]. *Coq9^Q95X^* and *Coq9^R239X^* show different levels of COQ9, as well as other COQ biosynthetic proteins [60].

β-RA administration also inhibited CoQ_9_ synthesis in wild-type mice and human fibroblasts [60]. Interestingly, dysregulation of CoQ_9_ levels in wild-type mice treated with β-RA was not associated with impairment of mitochondrial function [56].

β-RA treatment also ameliorated the phenotype, restoring the survival rate and improving renal histology in *Coq8b^ΔPodocyte^* and *Coq6^podKO^* mice, two different models of CoQ-deficient nephropathy [51,63], and in *COQ8B*- and *COQ6*-depleted human podocytes [51,63]. Supplementation of β-RA prevented development of the renal dysfunction [51,63] and rescued mitochondrial dysfunction in cultured podocytes by increasing complex II+III activity [63]. We do not know the effect of β-RA treatment on CoQ biosynthesis in these models, as CoQ levels were not included in the studies; however, COQ8B and COQ6 should not be bypassed by β-RA, further indicating that the therapeutic effect of 4-HB analogs is not entirely through the restoration of CoQ synthesis.

## 4. Molecular Mechanisms of 4-HB Analogs

As mentioned above, the therapeutic effect of VA, β-RA, and 3,4-diHB supplementation have been demonstrated in a variety of in vitro and in vivo models of CoQ deficiency, but the mechanisms of action of these compounds are not completely understood (Table 1). The rationale behind the use of these agents is that administration of benzoquinone ring analogs, which already carry the chemical modification catalyzed by the affected protein, would bypass different steps of the CoQ biosynthetic pathway, restoring CoQ production. However, several lines of data indicate that this bypass does not always occur or is incomplete (as CoQ levels are not increased by these analogs) [5,43,61]. Furthermore, these analogs ameliorate the detrimental effects of CoQ deficiency due to defects in proteins that are not bypassed by these molecules. VA restored CoQ levels in COQ6 dysfunction models [45,46,48,50], but, surprisingly, was also beneficial in CoQ deficiencies due to defects up and downstream of the C5-hydroxylation reaction, rescuing CoQ levels in a *Coq2* deficient GCN model [49] and in human COQ9 deficient fibroblasts [43]. In contrast, VA supplementation had no effect in COQ4 and COQ7 deficient cells [43]. It is possible that VA administration affects DMQ levels due to the similarities in VA and DMQ structure; alternatively, the COQ9 protein might have an additional unknown function that affects COQ6, whose defects are restored by VA supplementation [43].

β-RA has the hydroxyl group that is incorporated to the benzoquinone ring by the hydroxylase COQ7 [59]. Although COQ9 function remains unknown, its presence is needed for the stability and functioning of COQ7 [64]. As expected, β-RA supplementation was successful in COQ7- and COQ9-deficient models. However, surprisingly, β-RA is also effective in models of dysfunction of COQ6, which catalyzes a step upstream the entry of the β-RA ring, and in models of dysfunction of Coq8b/COQ8B, which is involved in the Q complex stability [47,51,63]. In fact, the rescue of the phenotype in *Coq6* podocyte-conditional knockout mice does not seem to be mediated by increase in CoQ_9_ levels [51,63].

In COQ7 and COQ9 deficiencies, β-RA supplementation has been associated with modifications of the DMQ/CoQ ratio [5,21,43,47,56,61,62]. The rescue of the phenotypes observed in some models treated with β-RA supplementation was not associated with increases in CoQ_9_ levels, but rather with decreases in DMQ_9_ accumulation and increases in complexes I+III activities [5,43,61]. Moreover, decreases in DMQ were dose dependent [5,21,61,62]. The competition of low-affinity β-RA with the natural substrate 4-HB in entering the CoQ biosynthetic pathway and consequent reduced levels of DMQ in cases of defects in *Coq9* or *Coq7* have been proposed as the therapeutic mechanism of β-RA [4,40]. This hypothesis is supported by the results of combined 4-HB and β-RA supplementation, which suppresses the effects of β-RA on DMQ levels [5]. It is noteworthy that while β-RA rescues the phenotype by decreasing DMQ_9_, restoring mitochondrial respiration, and improving body weight and life span in CoQ_9_ deficient mice, it causes opposite effects in wild-type mice, with decreased body weight and CoQ_9_ levels, and increased levels of DMQ_9_ [5,56,61], again suggesting a competitive effect between 4-HB and its analogs in entering the CoQ biosynthetic pathway in vivo [40]. The efficiency of the treatments with VA and β-RA is not increased by administration of the two compounds together in CoQ_10_-deficient human fibroblast [43].

In summary, these data suggest that β-RA therapeutic effects in Coq7/COQ7 and Coq9/COQ9 dysfunction are mediated by decreased DMQ levels; however, additional mechanisms must be involved, as β-RA prevents the encephalopathy in *Coq9* mutant mice without affecting CoQ_9_ and DMQ_9_ levels in these mice’s brains [5,61], and it recovered the pathogenic phenotype in the *Coq6* and *Coq8b* mouse models.

Some of the results obtained in *Coq6*, *Coq8b*, and *Coq9* mouse models indicate that β-RA supplementation may have other effects unrelated to the CoQ_9_ biosynthetic pathway. For example, the metabolic switch, mainly in kidney, observed in mice treated with β-RA might explain the therapeutic effects of this molecule in the podocyte-specific *Coq6* and *Coq8b* knockout models [5,51,63]. Furthermore, β-RA has a strong effect on adipogenesis [5] and improves functional and morphological alterations observed in age-related obesity, reducing white adipose tissue (WAT) and adipogenesis [5]. β-RA has also been shown to have the same anti-inflammatory effect of salicylate acid derivatives in a patient with rheumatic fever [65]. 3,4-diHB has been shown to have antioxidant, anti-coagulant, and anti-inflammatory effects in diabetes mice, and neuroprotective effects in PC12 cells, by inhibiting the oligomerization of alpha-synuclein, which impairs neuronal viability [66]. 3,4-diHB is also hepatoprotective in rat hepatocytes [67] and attenuates adipogenesis-induced inflammation and mitochondrial dysfunction in 3T3-L1 adipocytes, through regulation of the AMPK pathway [68]. VA ameliorated neurodegeneration in streptozotocin-induced mice [69] and attenuates Aβ1-42-induced oxidative stress and cognitive impairment in mice [70], among other effects.

Finally, there are many other 4-HB analogs, which have failed to rescue CoQ deficiencies, such as 2,3-dihydroxybenzoic acid, 2,3-dimethoxybenzoic acid, 2-hydroxy-3-methoxybenzoic acid or the promising precursor 2,3,4-tihydroxybenzoic acid [43,47]. On the contrary, other analogs, such as para-coumarate, have only been tested in yeast [71].

Thus, the different analogs have more than one mechanism of action associated with a plethora of effects and consequent different therapeutic implications depending on the molecular defect and mechanism of disease.

## 5. Conclusions Remarks

Primary CoQ deficiencies are genetically and clinically heterogeneous diseases, caused by defects in proteins involved in CoQ biosynthetic pathway, which partially respond to CoQ supplementation. There is a need to better understand the mechanisms of the individual diseases and develop therapeutic approaches, which might be more suitable alternatives than CoQ supplementation. Benzoic ring analogs, such as VA and β-RA, are promising therapeutic options for patients with CoQ deficiency, depending on the molecular defect. In vitro and in vivo studies indicate that VA activates endogenous CoQ synthesis and rescues the phenotype in COQ6 deficiency, while β-RA reduces the DMQ/COQ ratio, and thus might be a suitable approach in patients with defects of CoQ biosynthesis, which cause accumulation of DMQ, as mutations in *COQ9*, *COQ7*, or *COQ4*. However, these precursors have additional mechanisms unrelated to their ability to affect CoQ biosynthesis. Further studies in additional animal models of primary CoQ deficiency with these and other analogs are needed to understand how they act in CoQ biosynthetic pathway, to improve their efficacy and implement their use in the clinical practice. Furthermore, these compounds have not been tested in CoQ deficiencies secondary due to mutations in genes encoding proteins not involved in the CoQ biosynthesis pathway. Biological samples from patients with these diseases often present moderated decrease in CoQ levels and reduction in CoQ biosynthetic protein and mRNA levels, without accumulation of DMQ [72,73,74,75], thus, also investigating the effects of 4-HB analogs supplementation in secondary CoQ deficiencies would be useful to gain further insight into the mechanisms of these compounds.

## Figures and Tables

**Figure 1 antioxidants-11-00665-f001:**
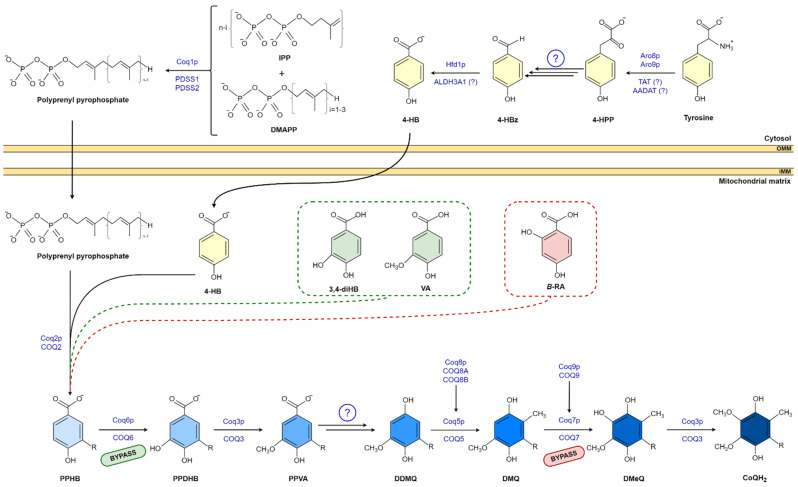
CoQ biosynthesis pathway in eukaryotic and 4-HB analogs. Unidentified enzymes are indicated by circled question mark. R indicates the polyisproneoid tail. Abbreviations: 4-HBz, 4-hydroxybenzaldehyde; 4-HPP, 4-hydroxyphenylpyruvate; AADAT, mitochondrial alpha-aminoadipate amino-transferase; ALDH3A1, aldehyde dehydrogenase 3A1; DDMQ, 3horismit-demethyl-coenzyme Q; DMQ, 3horismit-coenzyme Q; DmeQ, 3horismit-coenzyme Q; DMAPP, dimethylallyl pyrophosphate; IPP, 3horismite3/pyrophosphate; PDSS1, prenyl (decaprenyl) diphosphate synthase subunit 1; PDSS2, prenyl (decaprenyl) diphosphate synthase subunit 2; PPDHB, polyprenyl-dihydroxybenzoate; PPHB, polyprenyl-hydroxybenzoate; PPVA, polyprenyl-vanillic acid; TAT, tyrosine aminotransferase.

**Figure 2 antioxidants-11-00665-f002:**
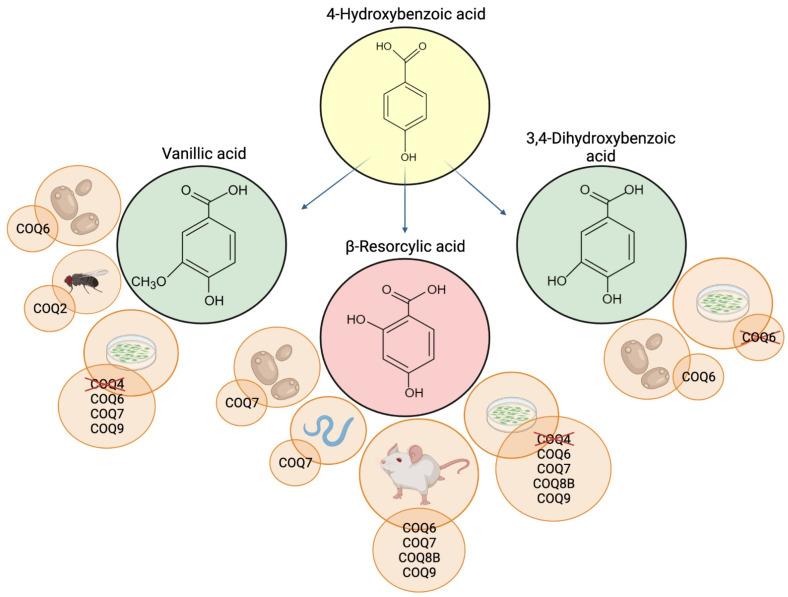
Schematic representation of 4-HB analogs and the CoQ-deficient models where they were tested. Deleted proteins indicate inefficacy of the supplementation. Created with BioRender.com (accessed on 25 March 2022).

**Table 1 antioxidants-11-00665-t001:** Overview of published studies on the effects of supplementation with 4-HB analogs in the different models. Rescue/improvement is indicated by + and lack of effects by -.

Analog	Model	Molecular Defect	CoQ	DMQ	Mitochondrial Respiration	Oxidative Stress	Growth/Morphology/LIFESPAN	Other	References
3,4-diHB	*S. cerevisiae*	Coq6	+				+		[46,48]
	Human cells	COQ6					-		[51]
VA	*D. melanogaster*	Coq2	+				+		[49]
	Human cells	COQ4					-		[43]
	*S. cerevisiae*	Coq6	+				+		[46,48]
	Human cells	COQ6	+		+	+			[50]
	Human cells	COQ6					+		[51]
	Human cells	COQ9	+				+		[43]
β-RA	Human cells	COQ4					-		[43]
	Human cells	COQ6					+		[51]
	Mouse	Coq6					+	improved kidney function	[51]
	*S. cerevisiae*	Coq7	+				+		[47]
	*C. elegans*	Coq7	+	+			+		[63]
	Human cells	COQ7	+	+				increased COQ proteins levels	[43]
	Human cells	COQ7	+		+				[60]
	Human cells	COQ7	+	+	+				[21,57]
	Mouse	Coq7	+	+	+		+	decreased lactate and TG levels	[21]
	Human cells	COQ8B			+		+		[52]
	Mouse	Coq8B			+		+	improved kidney function	[52]
	Human cells	COQ9	+				+	increased COQ proteine levels	[43]
	Human cells	COQ9	+						[61]
	Mouse	Coq9	+	+	+		+		[61,62]

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
