# Peer review of "Mechanisms and Therapeutic Effects of Benzoquinone Ring Analogs in Primary CoQ Deficiencies"

_antioxidants, 2022, doi:10.3390/antiox11040665_

Round 1

Reviewer 1 Report

This is a very well written review paper outlining the use of  4-hydroxybenzoic acid derivatives in the treatment of primary CoQ10 deficiencies.

I only have a few comments, please see below:

In the introduction there should be some mention of the use of CoQ10 in maintaining the acidity of the lysosomal lumen which is required for the functioning of the organelle. A recent paper by Heaton  R et al 2021 outlines this connection.

2. The authors indicate the limitations of CoQ10 supplementation in treating primary CoQ10 deficiencies, but don`t really provide a clear explanation. What about the limited transport of CoQ10 across the blood brain barrier. A recent paper by Wainwright et al., 2021 outlines this parameter in detail and should be included.

3. A schematic diagram outlining how these derivatives ameliorate the  oxidative stress, mito dysfunction etc associated with primary CoQ10 would complement the text.

4. Can these 4-hydroxybenozoate derivates be employed in  the treatment of 2ndary CoQ10 deficiencies?

Author Response

Thank you for your thoughtful and helpful comments about our work. We have carefully reviewed all the recommendations and have revised the manuscript. Our point-by-point responses to the comments are listed below.

  1. In the introduction there should be some mention of the use of CoQ10 in maintaining the acidity of the lysosomal lumen which is required for the functioning of the organelle. A recent paper by Heaton R et al 2021 outlines this connection.

We added the following sentence to the Introduction: “CoQ is also required for the maintenance of the lysosomal lumen acidity, necessary for the correct function of this organelle [6]”.

[6]. Heaton, R.A.; Heales, S.; Rahman, K.; Sexton, D.W.; Hargreaves, I. The effect of cellular coenzyme q10 deficiency on lysosomal acidification. J. Clin. Med. 2020, 9, 1–14, doi:10.3390/jcm9061923.”

  1. The authors indicate the limitations of CoQ10 supplementation in treating primary CoQ10 deficiencies, but don`t really provide a clear explanation. What about the limited transport of CoQ10 across the blood brain barrier. A recent paper by Wainwright et al., 2021 outlines this parameter in detail and should be included.

We added the following sentence to explain why CoQ supplementation is not always an effective therapy: “The low bioavailability of CoQ and its inability to cross the blood–brain barrier (BBB) indicates the need of and alternative therapeutic approaches are needed [9], especially in patients with neurological involvement [10].”

[9]. Wainwright, L.; Hargreaves, I.P.; Georgian, A.R.; Turner, C.; Dalton, R.N.; Abbott, N.J.; Heales, S.J.R.; Preston, J.E. CoQ10 deficient endothelial cell culture model for the investigation of CoQ10 blood–brain barrier transport. J. Clin. Med. 2020, 9, 1–21, doi:10.3390/jcm9103236.”

  1. A schematic diagram outlining how these derivatives ameliorate the oxidative stress, mito dysfunction etc associated with primary CoQ10 would complement the text.

We added a table summarizing the findings of all the studies on the derivative and their effects.

  1. Can these 4-hydroxybenozoate derivates be employed in the treatment of 2ndary CoQ10 deficiencies?

As far as we know, these compounds have never been used in secondary CoQ deficiencies. We added the following paragraph to the Conclusions: “These compounds have not been tested in CoQ deficiencies secondary due to mutations in genes encoding protein not involved in the CoQ biosynthesis pathway. Biological samples from patients with these diseases often present moderated decrease of CoQ levels, and reduction of CoQ biosynthetic protein and mRNA levels, without accumulation of DMQ, thus, also investigating the effects of 4-HB analogues supplementation in secondary CoQ deficiencies would be useful to gain further insight into the mechanisms of these compounds.

Reviewer 2 Report

The review is very well written and organized and updates the latest findings on the therapeutic potential of benzoquinone ring analogues in primary CoQ deficiencies. The figures summarize and outline the information provided.

Author Response

We thank the reviewer for the positive comments on our manuscript.